# Trends and determinants of acute respiratory infection symptoms among under-five children in Cambodia: Analysis of 2000 to 2014 Cambodia demographic and health surveys

**Samnang Um**[1]*, **Daraden Vang**[1], **Punleak Pin**[2], **Darapheak Chau**[1]

**1** National Institute of Public Health (NIPH), Phnom Penh, Cambodia, **2** Doctoral specialization in Pneumology, Faculty of Medicine at the University of Health Sciences (UHS) in Phnom Penh, Phnom Penh, Cambodia

* umsamnang56@gmail.com

**Data Availability Statement:** Our study used the 2000, 2005, 2010, and 2014 Cambodia Demographic and Health Survey datasets and are

## Abstract

Acute Respiratory Infections (ARIs) are the leading cause of mortality and morbidity among children under 5 years old and about 1.3 million annually worldwide. Account for 33% of deaths among children under 5 years that occurred in developing countries. In Cambodia, ARIs prevalence in children under 5 years old was 20% in 2000, and 6% in 2014. Hence, the aimed to describe the trends of ARI symptoms among children aged 0–59 months over time using the 2000, 2005, 2010, and 2014 Cambodia Demographic and Health Survey (CDHS) and determined the relationships between socio-demographic, behavioral, and environmental factors with **ARI symptoms.** We analyzed existing children's data from 2000, 2005, 2010 and 2014 of Cambodia Demographic and Health Survey (CDHS) that used a two-stage stratified cluster sampling design. We limited our analysis to children born in the last five years prior to the surveys, alive and living in households during interview time. Data were pooled across the four survey years for 29,171 children aged 0–59 months. All statistics were carried out using STATA V16, and survey weights were taken into account for the survey design of the CDHS. We used multiple logistic regression to determine the main predictors of **ARI symptoms** among children under 5 years. ARI symptoms in the previous two weeks in children aged 0–59 months in Cambodia were 19.9% in 2000 to 8.6% in 2005 to 6.4% in 2010, and 5.5% in 2014. Factors independently associated with increased odds of ARI symptoms were children ages 6–11 months with adjusted odds ratio [AOR = 1.91; 95% CI: 1.53–2.38], 12–23 months [AOR = 1.79; 95% CI: 1.46–2.20], and 24–35 months [AOR = 1.41; 95% CI: 1.13–1.76], smoking mother [AOR = 1.61; 95% CI: 1.27–2.05], and using non-improved toilets in households [AOR = 1.20; 95% CI: 0.99–1.46]. However, the following factors were found to be associated with decreased odds of having ARI symptoms: Mothers with higher education [AOR = 0.45; 95% CI: 0.21–0.94], breastfeeding children [AOR = 0.87; 95% CI: 0.77–0.98], and children born into richest wealth quantile [AOR = 0.73; 95% CI: 0.56–0.95], respectively. Survey 2005 [AOR = 0.36; 95% CI: 0.31–0.42],

publicly available to the DHS website at https://www.dhsprogram.com/data/dataset_admin.

**Funding:** The authors received no specific funding for this work.

**Competing interests:** The authors have declared that no competing interests exist.

2010 [AOR = 0.27; 95% CI: 0.22–0.33], 2014 [AOR = 0.24; 95% CI: 0.19–0.30]. The trends of ARI symptoms among children under five in Cambodia significantly decreased from 2000–2014. Smoking mothers, young children ages (0–35 months), and using non-improved toilet in household are factors that independently increased the likelihood that children would develop ARI symptoms. Inversely, factors were found to be associated with decreased odds of having ARI symptoms: Mothers with higher education, breastfeeding children, and children born into the richest wealth quantile and Survey years. Therefore, government and child family programs must promote maternal education, particularly infant breastfeeding. The government ought to support maternal education and infant breastfeeding in the interest of early childhood care.

## Introduction

Acute respiratory infections (ARIs) are public health concern as it has been a leading cause of mortality and morbidity among children under 5 years worldwide [1]. It is estimated that five million children under 5 years die of ARIs worldwide in 2020. All of these deaths could be avoided if patients had access to more affordable health, sanitation, and hygienic interventions [1]. ARIs is defined as an infection of the airways, which includes the upper and lower respiratory tracts, brought on by a variety of pathogens including bacteria, viruses, fungi, and parasites. Most children diagnosed with ARIs, but not those having blocked nose exhibit one or more signs and symptoms such as coughing, difficulty breathing or dyspnea, or tachypnea [2–4]. Globally,1.3 million under 5 years die from ARIs annually [5]. ARIs account for 33% of deaths among children under 5 years of age that occurred in developing countries, particularly in Southeast Asia [6]. According to World Health Organization (WHO), an estimated 3.5% of the global disease burden is caused by ARI, and is responsible for between 30% to 50% of all pediatric outpatient visits and more than 30% of pediatric admissions in low and middle-income countries [1]. In Cambodia, the proportion of children under 5 years who reported ARI symptoms in the 2 weeks preceding the survey were at 20% in 2000, and significantly decreased to at 6% in 2014 [7, 8]. Due to ARI symptoms, children between 37% in 2000 and 69% in 2014 were brought to a medical facility in Cambodia [7–9]. Nearly 83% of children with symptoms of ARIs received antibiotics in 2014 [7–9]. In 2017, Children under 5 years of age deaths due to respiratory infections for Cambodia was 0.14% [10], making the leading causes morbidity and mortality among children aged under 5 years in Cambodia [7].

Previous research conducted in Zambia showed that mothers had completed at least secondary education was less likely to have ARI symptoms among their children (AOR = 0.30) compared with no education [11]. Similar studied at Indonesia found that maternal who had high education, the less the possibility of experiencing ARI symptoms among their children [12]. Underweight children were 1.5 times more likely to have ARI symptoms as compared to normal weight [11]. The risk of having ARI symptoms was almost 2.7 times higher among children living in households using electricity compared to those using biomass fuels like charcoal and wood (AOR = 2.67) [11]. Female children had a lower probability of experiencing ARI symptoms than male ones (OR = 0.89) [12]. Children under the age of one had a significant effect on the susceptibility of ARI symptoms [11, 13, 14]. Another study in India found higher prevalence of ARI symptoms among children living in an Urban areas compared to rural children [15]. Children from low-income families who lack access to healthcare services are higher probability develop ARI symptoms [13–16]. ARI symptoms were reported by 10% of children

whose moms smoke compared to 5% of children whose mothers do not smoke. In addition, 7% of children in families with the lowest levels of wealth had ARI symptoms, compared to 4% of children in families with the highest levels of wealth [7]. Measles vaccinated children was associated with reduction in ARIs cases by 15–30% in India and Pakistan [17]. Also, studied conducted in rural northern Bangladesh observed that the use of unimproved toilet facilities increased the risk of childhood acute respiratory infection by 31% [18]. Another recent study conducted in Myanmar observed children in the household with unimproved toilet facility were at significantly higher risk of suffering from cough and fever compared to households with improved toilets [19]. Despite, the global burden of children under 5 years morbidity and mortality attributable to ARIs, there have been limited data to evaluate the trends and define the potential factors associated with the presence of **ARI symptoms** among children aged 0–59 months. Our study thus, aimed to describe the trends of ARI symptoms among children aged 0–59 months over time using the 2000, 2005, 2010, and 2014 Cambodian Demographic and Health Survey (CHDS) data and determined the relationships between socio-demographic, behavioral, and environmental factors with **ARI symptoms** in children under 5 years in Cambodia.

## Methods

### Ethnics statement

The data used in this study were extracted from CDHS 2000, 2005, 2010 2014 dataset, which are publicly available with all personal identifiers of study participants removed. Permission to analyze the data was granted through registering with the DHS program website [25]. Informed consent was obtained from all participants before data collection. The data collection tools and procedures for CDHS was approved by the Cambodia National Ethics Committee for Health Research and the Institutional Review Board (IRB) of ICF in Rockville, Maryland, USA.

### Data source

We used existing children's data from the 2000, 2005, 2010, and 2014 Cambodia Demographic and Health Survey (CDHS) datasets. The CDHS is a population-based household survey that is regularly carried out every five years to collect the data on demographic and health information from nationally representative samples. The two-stage stratified cluster sampling methods was used to collect the samples from all provinces that are divided into sampling domains. They were further divided into sampling strata between urban and rural. In the first stage, cluster, or enumeration areas (EAs), that represents the entire country (urban and rural) are randomly selected from the sampling frame using probability proportional (PPS) to cluster size. In the second stage, a complete listing of households was selected from each cluster chosen using an equal probability systematic sampling, and then interviews with women between the aged 15–49 years who were born in the five years preceding the survey in the full list selected households. Details of CDHS design and data collection procedures have been described elsewhere [7–9]. We limited our analysis to children born in the last five years prior to the surveys, alive and living with their mothers or caregivers during interview time.

### Measurements

**Outcome variable.** In CDHS, ARI symptoms among children under 5 years are defined as the occurrence of cough accompanied by short, rapid breathing in the two weeks preceding

the survey. Our outcome variable, ARI symptoms among children under 5 years was a binary variable, coded as **1** for the presence of **ARI symptoms** and **0** otherwise.

**Independent variables.** Mother's **Age** was categorized into 15–19 (reference group), 20–24, 25–29, 30–34, 35–39, and 40–49. Mother's **Education** was coded into ordinal level variable with no education (reference group), primary, secondary, and higher. Mother's **Employment** was coded into a dichotomous variable was not working vs working and Mother's **Smoking** was coded into a dichotomous variable was smoker vs non-smoker. Child's **Age** in months was coded as 0–5 (reference group), 6–11, 12–23, 24–35, 36–47, and 48–59. **Birth order** was coded into ordinal variable with 1st child (reference group), 2nd-3rd children, and at less 4 children. Child's **Sex**, and **Weight's at birth** in kilogram were coded into dichotomous variable was <2.5kg vs > 2.5kg. **BCG vaccination status** was coded in to dichotomous variable with Incomplete vs Completed. **Intake vitamin A last 6 months, Breastfeeding status. Places of delivery** was categorized: Public facilities (reference group), Private facilities and at home). **Households wealth quintile** were calculated scores based on household assets (television, bicycle/car, size of agricultural land, quantity of livestock), and dwelling characteristics (sources of drinking water, sanitation facilities, and materials used for constructing houses) using principal component analysis (PCA), and the scores given into five categories of wealth quintile (poorest, poorer, medium, richer, and richest) each comprising 20% of the population [7–9]. Source of drinking water was coded as **Improved drinking water** included rainwater, piped into the home, piped into the yard or plot, public taps or standpipes, tubed wells or boreholes, and protected wells and springs another source as non-improved. **Toilet facilities** that have ventilated/improved latrines or other types of toilets are considered improved, whereas those that do not have any toilets are considered unimproved. **Residence** areas (urban vs rural). **CDHS survey years** was categorized 2000 (reference group), 2005, 2010, and 2014.

## Statistical analysis

All statistical analysis performed by STATA version 16 (Stata Corp 2019, College Station, TX) The complex survey design was declared to be taken into account using the STATA command "survey" package, and all estimations were carried out using the survey-specific command "svy" using the standard sampling weight (v005/1,000,000), clustering, and stratification variables that were provided by DHS. We estimate of overall trends in ARI symptoms prevalence among children aged 0–59 months over time. Bivariate chi-square tests were used to associations between the independent variables of interest, such as the socio-demographics, behavioral and environmental factors, and **ARI status**. Variables associated with the outcome variable with a significance level of p-value **0.10** or background variables mother's age, child's age, place of residence, and survey years were included in the multiple logistic regression analysis [20, 21]. Unadjusted logistic regression was used to determine the magnitude effect of associations between ARI symptoms socio-demographics behavioral and environmental factors reported as odds ratios (OR) with 95% confidence intervals (CI). Then, multiple logistic regression was used to assess independent associations, reported as adjusted odds ratios (AOR), with ARIs symptoms after adjusting for other independent variables included in the model. Multicollinearity was checked for some original variables including women's age and, education, and wealth index.

## Results

### Trends of ARI symptoms among children age 0–59 months

Table 1 presents the distribution of ARI symptoms among children under five across the survey year. A total of 29,171 children under 5 years old were analyzed, included 7,284 in 2000,

**Table 1. Trends ARI symptoms of Cambodia children aged between 0–59 months, CDHS 2000 to 2014.**

| Variable | 2000 (n = 7,284) | | 2005 (n = 7,201) | | 2010 (n = 7,730) | | 2014 (n = 6,956) | | P-value |
|---|---|---|---|---|---|---|---|---|---|
| | % | 95%CI | % | 95%CI | % | 95%CI | % | 95%CI | |
| **ARI symptoms past two weeks** | | | | | | | | | |
| Yes | 19.9 | [18.4–21.5] | 8.6 | [7.7–9.6] | 6.4 | [5.6–7.4] | 5.5 | [4.8–6.3] | <0.001 |
| No | 80.1 | [78.5–81.6] | 91.4 | [90.4–92.3] | 93.6 | [92.6–94.4] | 94.5 | [93.7–95.2] | |

7,201 in 2005, 7,730 in 2010, and 6,956 in 2014 respectively. Overall, trend of children age 0–59 months having ARI symptoms in the previous two weeks were significantly declined over time from 19.9% [95% CI: 18.4–21.5] in 2000 to 8.6% [95% CI: 7.7–9.6] in 2005 to 6.4% [95% CI: 5.6–7.4] in 2010 and decreased to 5.5% [95% CI: 4.8–6.3%] in 2014 (p value < 0.001). When, stratified by age, the ARIs trend prevalence peaked at around 30% among children aged 5–20 months. However, this trend among this age group gradually declined to less than 20% in 2005, 10% in 2010, and 2014. The ARI prevalence varied from 2% to 12% among different age groups under 5 years in 2010 and 2014 (**Fig 1**).

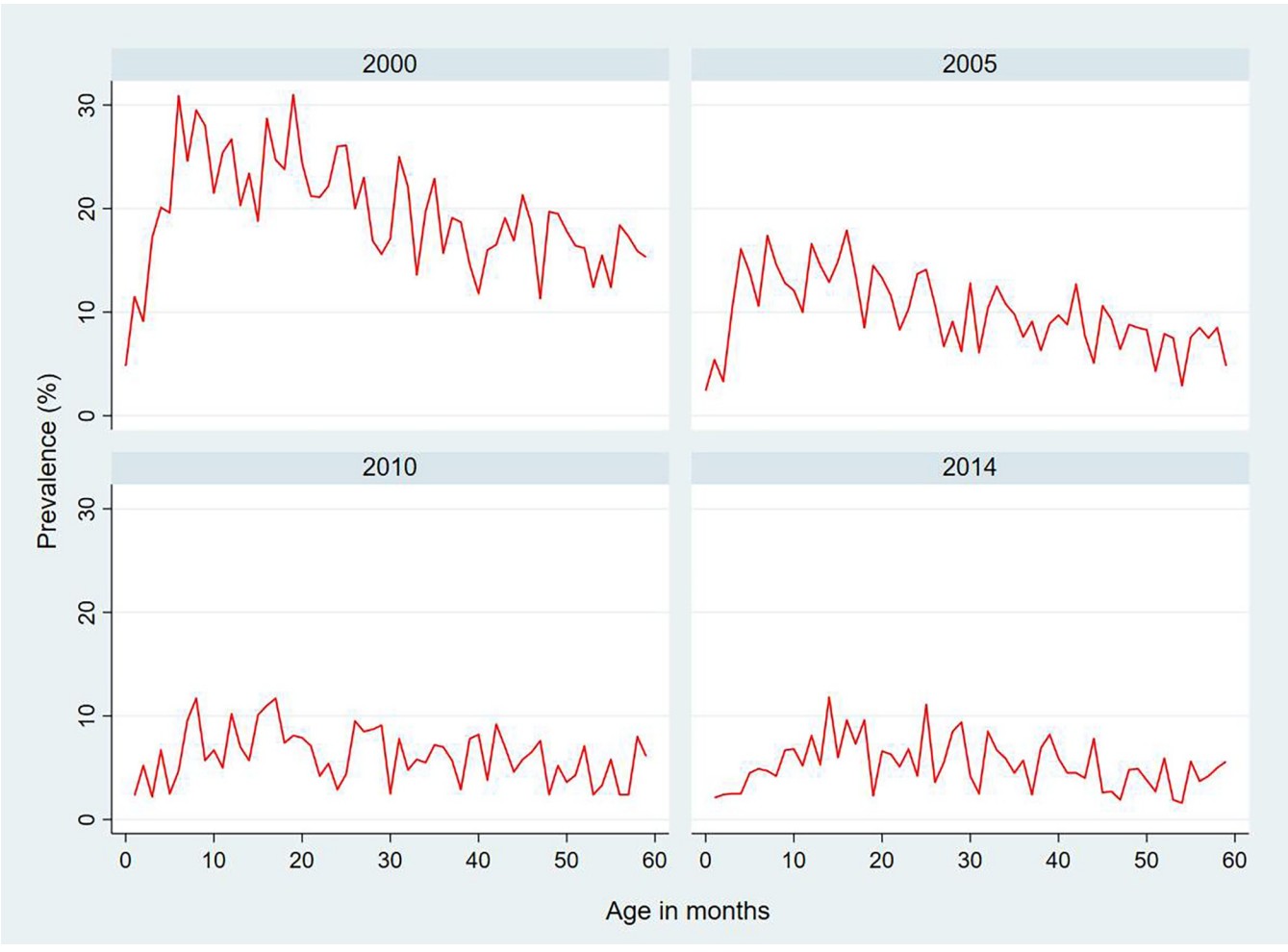

**Fig 1. Trends of children aged 0–59 months with ARI symptoms by survey years, CDHS 2000 to 2014.**

## Descriptive and association with ARI symptoms among children ages 0–59 months in bivariate analysis

Describes study population (**Table 2**). Mother's aged 15–19 years old were 22.9%, 26.1%, 24.7%, and 26.3% from 2000, 2005, 2010 and 2014. While mothers have no education 37.4% in 2000, 28.8% in 2005, 21.6% in 2010, only 12.5% in 2014. Most of the mothers was smoking were 32.6%, 36.8%, 16.9% and 13.7%.

In bivariate analysis the proportion of ARI symptoms among U5 children born from illiterate mothers was significantly higher than that among children born from educated mothers across the three latest survey (4.4%, 2.5%, and 2.7%, respectively). Smoking mothers were more likely to have children with ARI symptoms than non-smoking mothers (26% in 2000 at p value 0.023, 12.3% in 2005 at p value 0.025, 15.4% in 2010 at p value 0.001, and 17.5% at p value 0.001). ARI symptoms were related to children's ages with a higher prevalence in the age group of 6 to 11 months (6–11 months, 27.4% in 2000, 11.0% in 2005, 8% in 2010, and 6% in 2014; 12–23 months, 24%, 11%, 8.6%, and 7.5%, respectively; p values 0.001 in 2000–2010 and 0.014 in 2014). Children born and raised in well-off families were less likely to have ARI symptoms. This dose-response relationship was significant in 2005 and 2014 data with a p-value of <0.001 in 2005 and 0.03 in 2010. In addition, families using non-improved toilets were less likely to have ARI symptoms for their children. This dose-response relationship was significant in 2005, 2010 and 2014 with a p-value of <0.05 (**Table 2**).

## Determinants of ARI symptoms in multiple logistic analysis

In the final multiple logistic model (**Table 3**), Factors independently associated with increased odds of ARI symptoms among children aged 0–59 months included, children's ages, smoking mothers, household using non-improved toilets. However, factors were found to be associated with decreased odds of having ARI symptoms: educated months, breastfeed children, children born and raise well economic families, and survey years.

Older children aged 6–11 months [AOR = 1.91; 95% CI: 1.53–2.38], 12–23 months [AOR = 1.79; 95% CI: 1.46–2.20], 24–35 months [AOR = 1.41; 95% CI: 1.13–1.76] were more likely to have ARI symptoms compared to young children aged 0–5 months. Maternal smoking status had a significant effect on children's ARI symptoms [AOR = 1.61; 95% CI: 1.27–2.05]. In addition, children from households with non-improved toilets facility were more likely to have ARI symptoms but low statistically significant compared to those household with improved latrine [AOR = 1.20; 95% CI:0.99–1.46; P = 0.064]. On the contrary, the factors negatively associated with having ARI symptoms among children under 5 years included mother with higher education [AOR = 0.45; 95% CI: 0.21–0.94]. Child breastfeeding [AOR = 0.87; 95% CI: 0.77–0.98]. Being born into richest wealth quantile [AOR = 0.73; 95% CI: 0.56–0.95]. Survey years in 2005 [AOR = 0.36; 95% CI: 0.31–0.42], 2010 [AOR = 0.27; 95% CI: 0.22–0.33], 2014 [AOR = 0.24; 95% CI: 0.19–0.30] (**Table 3**).

## Discussion

ARI prevalence among children under 5 years in Cambodia significantly decreased over time from 19.5% in 2000 to 5.2% in 2014. The reduction of the disease burden corresponded with global efforts to reduce under 5 mortality and due to ARI and diarrhea [22–24]. It might be attributed to the efforts and initiative of the Cambodian National Immunization Program's efforts to replace the DPT vaccine with a tetravalent vaccine that includes DPT and the Hib vaccine and a pentavalent vaccine that includes the DPT, Hib, and hepatitis B vaccine are responsible for the decrease in the prevalence of ARI that was observed (HepB). The HepB

**Table 2. Proportion of children age 0–59 months who have ARI symptoms over survey years by characteristics in bivariate chi-square analysis, CDHS 2000 to 2014.**

| Variables | 2000(n = 7,284) | | | 2005 (n = 7,201) | | | 2010(n = 7,730) | | | 2014(n = 6,956) | | |
|---|---|---|---|---|---|---|---|---|---|---|---|---|
| | Total | % ARI | P value | Total | % ARI | P value | Total | % ARI | P value | Total | %ARI | P value |
| **Mother age in years** | | | | | | | | | | | | |
| < 19 | 85 | 25 | 0.791 | 89 | 13.2 | 0.182 | 90 | 10.8 | 0.514 | 92 | 2.1 | 0.637 |
| 19–24 | 1,208 | 19.7 | | 1,905 | 9.4 | | 1,821 | 7.1 | | 1,814 | 5.7 | |
| 25–29 | 1,938 | 18.8 | | 1,808 | 7.6 | | 2,727 | 6.6 | | 2,148 | 5.0 | |
| 30–34 | 1,792 | 20.2 | | 1,508 | 8.8 | | 1,539 | 5.5 | | 1,843 | 5.6 | |
| 35–39 | 1,367 | 20.5 | | 1,136 | 9.4 | | 919 | 5.9 | | 701 | 6.7 | |
| 40+ | 893 | 20.3 | | 755 | 6.5 | | 635 | 6.5 | | 358 | 5.5 | |
| **Mother educational** | | | | | | | | | | | | |
| No education | 2,354 | 19.8 | 0.692 | 1,720 | 9.3 | **0.048** | 1,414 | 7.2 | **0.026** | 956 | 5.4 | **0.025** |
| Primary | 3,888 | 20.1 | | 4,242 | 9.1 | | 4,374 | 7.0 | | 3,656 | 6.5 | |
| Secondary | 1,029 | 19.5 | | 1,198 | 6.1 | | 1,822 | 4.8 | | 2,155 | 4.2 | |
| Higher | 12 | 4.4 | | 42 | 0.0 | | 120 | 2.5 | | 188 | 2.7 | |
| **Mother working** | | | | | | | | | | | | |
| Not working | 2,117 | 19.9 | 0.962 | 2,948 | 9.1 | 0.305 | 2,623 | 5.7 | 0.076 | 2,515 | 4.5 | 0.048 |
| Working | 5,166 | 19.8 | | 4,253 | 8.2 | | 5,108 | 6.8 | | 4,437 | 6.1 | |
| **Mother smokes** | | | | | | | | | | | | |
| Non-Smoker | 6,842 | 19.5 | **0.023** | 6,859 | 8.4 | **0.025** | 7,518 | 6.2 | **<0.001** | 6,759 | 5.4 | **0.028** |
| Smoker | 440 | 25.8 | | 342 | 12.3 | | 212 | 15.4 | | 196 | 10.3 | |
| **Child's age in months** | | | | | | | | | | | | |
| ≤ 5 | 810 | 14.6 | **<0.001** | 742 | 8.7 | **<0.001** | 711 | 3.4 | **<0.001** | 729 | 3.0 | **0.014** |
| 6–11 | 785 | 27.4 | | 770 | 11.0 | | 824 | 7.7 | | 758 | 6.0 | |
| 12–23 | 1,248 | 23.9 | | 1,512 | 10.9 | | 1,606 | 8.6 | | 1,443 | 7.5 | |
| 24–35 | 1,368 | 21.1 | | 1,404 | 8.9 | | 1,596 | 7.3 | | 1,350 | 6.0 | |
| 36–47 | 1,530 | 17.7 | | 1,405 | 7.1 | | 1,507 | 6.3 | | 1,319 | 5.5 | |
| 48–59 | 1,542 | 16.5 | | 1,368 | 5.8 | | 1,487 | 4.1 | | 1,356 | 4.0 | |
| **Sex of child** | | | | | | | | | | | | |
| Male | 3,675 | 20.1 | 0.686 | 3,579 | 9.1 | 0.118 | 3,996 | 6.9 | 0.234 | 3,485 | 5.8 | 0.465 |
| Female | 3,609 | 19.6 | | 3,622 | 8.0 | | 3,734 | 5.9 | | 3,471 | 5.3 | |
| **Birth order** | | | | | | | | | | | | |
| 1 child | 1,390 | 19.1 | 0.021 | 1,995 | 9.2 | 0.518 | 2,650 | 6.7 | 0.187 | 2,704 | 4.7 | 0.155 |
| 2–3 | 2,657 | 18.5 | | 2,861 | 7.9 | | 3,356 | 5.8 | | 3,163 | 5.6 | |
| 4–5 | 1,619 | 23.1 | | 1,387 | 8.6 | | 1,114 | 8.1 | | 799 | 6.8 | |
| 6+ | 1,618 | 19.6 | | 958 | 9.3 | | 610 | 5.5 | | 290 | 8.1 | |
| **Birth weight (kg)** | | | | | | | | | | | | |
| ≥2.5kg | 1,074 | 22.5 | 0.369 | 2,490 | 6.4 | 0.004 | 4,933 | 6.7 | 0.457 | 5,518 | 5.5 | 0.900 |
| <2.5kg | 6,210 | 19.4 | | 4,711 | 9.7 | | 2,797 | 6.0 | | 1,438 | 5.6 | |
| **Place of delivery** | | | | | | | | | | | | |
| Public | 618 | 19.5 | 0.945 | 1,230 | 5.6 | 0.002 | 3,417 | 6.5 | 0.440 | 4,807 | 5.4 | 0.796 |
| Private | 129 | 22.0 | | 351 | 3.7 | | 780 | 5.0 | | 1,000 | 5.6 | |
| At Home | 6,520 | 19.9 | | 5,621 | 9.5 | | 3,533 | 6.7 | | 1,149 | 6.1 | |
| **Breastfeeding** | | | | | | | | | | | | |
| No | 2,901 | 20.3 | 0.583 | 3,140 | 6.8 | <0.001 | 3,877 | 6.9 | | 3,785 | 5.5 | 0.981 |
| Yes | 4,383 | 19.6 | | 4,061 | 9.9 | | 3,853 | 6.0 | | 3,171 | 5.5 | |
| **BCG-vaccinated** | | | | | | | | | | | | |
| None | 2,534 | 19.5 | 0.651 | 816 | 8.9 | 0.710 | 635 | 6.1 | 0.767 | 410 | 6.4 | 0.493 |
| Yes | 4,750 | 20.1 | | 6,385 | 8.5 | | 7,095 | 6.5 | | 6,546 | 5.5 | |

(*Continued*)

**Table 2.** (Continued)

| Variables | 2000(n = 7,284) | | | 2005 (n = 7,201) | | | 2010(n = 7,730) | | | 2014(n = 6,956) | | |
|---|---|---|---|---|---|---|---|---|---|---|---|---|
| | Total | % ARI | P value | Total | % ARI | P value | Total | % ARI | P value | Total | %ARI | P value |
| **Intake vitamin A last 6 months** | | | | | | | | | | | | |
| No | 5,165 | 20.1 | 0.707 | 4,823 | 8.8 | 0.524 | 2,765 | 5.9 | 0.333 | 2,536 | 4.4 | 0.018 |
| Yes | 2,081 | 19.6 | | 2,378 | 8.2 | | 4,966 | 6.7 | | 4,420 | 6.2 | |
| **Place of residence** | | | | | | | | | | | | |
| Urban | 967 | 19.4 | 0.707 | 1,003 | 5.3 | **<0.001** | 1,219 | 3.3 | **<0.001** | 987 | 5.6 | 0.944 |
| Rural | 6,317 | 19.9 | | 6,198 | 9.1 | | 6,512 | 7.0 | | 5,968 | 5.5 | |
| **Wealth index** | | | | | | | | | | | | |
| Poorest | 1,863 | 18.4 | 0.530 | 1,934 | 12.4 | **<0.001** | 2,031 | 7.9 | **0.003** | 1,678 | 7.0 | 0.205 |
| Poorer | 1,656 | 20.5 | | 1,635 | 9.8 | | 1,652 | 7.0 | | 1,394 | 5.3 | |
| Middle | 1,444 | 19.6 | | 1,268 | 8.4 | | 1,416 | 7.3 | | 1,327 | 5.1 | |
| Richer | 1,357 | 21.7 | | 1,167 | 6.5 | | 1,352 | 5.4 | | 1,205 | 5.5 | |
| Richest | 963 | 19.5 | | 1,197 | 3.0 | | 1,280 | 3.4 | | 1,351 | 4.4 | |
| **Drinking water** | | | | | | | | | | | | |
| Improved | 4,559 | 20.5 | 0.223 | 3,747 | 7.9 | 0.106 | 4,205 | 5.7 | 0.040 | 3,494 | 5.5 | 0.904 |
| Unimproved | 2,718 | 18.8 | | 3,454 | 9.3 | | 3,526 | 7.4 | | 3,462 | 5.6 | |
| **Toilet facility** | | | | | | | | | | | | |
| Improved | 753 | 17.2 | 0.136 | 1,558 | 4.4 | **<0.001** | 2,755 | 5.4 | 0.081 | 3,454 | 4.8 | **0.035** |
| Non-Improved | 6,529 | 20.2 | | 5,643 | 9.7 | | 4,975 | 7.0 | | 3,502 | 6.2 | |

vaccine is also given as part of the program in 2006 [7], either at birth or during the first clinical encounter. Additionally, from 40% in 2000 to 96% in 2014, more children received the BCG vaccine [7]. Children of smoking mother had 1.6 times higher risk of developing ARI symptoms compared to children of non-smoking mothers. The findings were consistent with those of few Sub-Saharan African studies [4, 25–27]. According to WHO, children who are exposed to parental smoking were more likely to have pneumonia and other respiratory infection diseases [28]. Particularly in areas where smoking and the use of firewood are common, parents and the community need to be educated about the risks that smoking has for children [6, 29]. ARI symptoms in young children (0–35 months) were common in our study than in older children (36–59 months). These results are in line with other studied [6, 13, 30]. The increased risk for ARI in this age group might be explained by the children's low immunity. The immune system appeared to be stronger at a later stage in older children after vaccination. In particularly in countries in Southeast Asia and sub-Saharan African, where health facilities and maternal healthcare education need to be improved, the factors were low rates of immunization in young children, low maternal literacy, and young mothers engaged in farming activities that prevent the care of young children [30, 31]. Additionally, compared to children living households with improved toilet facilities, children from households with unimproved toilet facilities were more likely to experience ARI symptoms. Improved sanitation facilities were found in a supporting multicounty WASH-intervention study to reduce the risk of children from fever by 13% and cough by 10%, according to a [32]. Experience ARI symptoms among children under 5 years were found some studies in Nigeria and Myanmar found that households lacking all three types of WASH facilities had higher odds of having cough, fever, and diarrhea [19, 33]. On the other hand, our finding on the association between child breastfeeding and the presence of ARI symptoms was inconsistent with findings of other studies [34, 35]. According to the current study, non-breastfed child was found to be more likely having experience of ARI symptoms compared breastfed children. In general, breastfeeding is more crucial

**Table 3. Factors independently associated with ARI symptoms among children aged 0–59 months in multiple logistic regression analysis, CDHS 2000 to 2014.**

| Variables | | 2000–2014 (N = 29,171) | | | 2000–2014 (N = 29,171) | | 2000–2014 (N = 29,115) | |
|---|---|---|---|---|---|---|---|---|
| | | Total | % ARI | P value | OR | 95% CI | AOR | 95% CI |
| **Mother age in year** | | | | | | | | |
| | < 19 | 356 | 12.6 | **0.002** | 1.0 | ref | 1.0 | ref |
| | 19–24 | 6,749 | 9.6 | | 0.74 | (0.51–1.07) | 0.79 | (0.54–1.16) |
| | 25–29 | 8,622 | 9.2 | | 0.70* | (0.48–1.01) | 0.73 | (0.49–1.07) |
| | 30–34 | 6,681 | 10.2 | | 0.79 | (0.55–1.14) | 0.74 | (0.50–1.11) |
| | 35–39 | 4,122 | 11.8 | | 0.93 | (0.64–1.36) | 0.77 | (0.50–1.19) |
| | 40+ | 2,642 | 11.0 | | 0.86 | (0.59–1.26) | 0.71 | (0.46–1.10) |
| **Mother working** | | | | | | | | |
| | Not working | 10,203 | 9.3 | **0.031** | 1.0 | ref | 1.0 | ref |
| | Working | 18,964 | 10.5 | | 1.14** | (1.01–1.28) | 1.03 | (0.91–1.16) |
| **Mother educational** | | | | | | | | |
| | No education | 6,443 | 12.1 | **<0.001** | 1.0 | ref | 1.0 | ref |
| | Primary | 16,161 | 10.6 | | 0.86** | (0.76–0.98) | 1.09 | (0.95–1.23) |
| | Secondary | 6,204 | 7.3 | | 0.57*** | (0.48–0.68) | 0.93 | (0.77–1.13) |
| | Higher | 363 | 2.4 | | 0.18*** | (0.09–0.37) | **0.45**** | **(0.21–0.94)** |
| **Mother smokes** | | | | | | | | |
| | Non-Smoke | 27,979 | 9.8 | **<0.001** | 1.0 | ref | 1.0 | ref |
| | Smoker | 1,191 | 17.5 | | 1.96*** | (1.58–2.43) | **1.61**** | **(1.27–2.05)** |
| **Child age in months** | | | | | | | | |
| | ≤ 5 | 2,992 | 7.6 | **<0.001** | 1.0 | ref | 1.0 | ref |
| | 6–11 | 3,137 | 13.0 | | 1.81*** | (1.46–2.25) | **1.91**** | **(1.53–2.38)** |
| | 12–23 | 5,810 | 12.2 | | 1.68*** | (1.39–2.04) | **1.79**** | **(1.46–2.20)** |
| | 24–35 | 5,719 | 10.7 | | 1.45*** | (1.18–1.77) | **1.41**** | **(1.13–1.76)** |
| | 36–47 | 5,761 | 9.3 | | 1.24** | (1.02–1.52) | 1.15 | (0.92–1.44) |
| | 48–59 | 5,753 | 7.8 | | 1.02 | (0.84–1.25) | 0.94 | (0.76–1.18) |
| **Sex of child** | | | | | | | | |
| | Male | 14,736 | 10.5 | 0.078 | 1.0 | ref | 1.0 | ref |
| | Female | 14,436 | 9.7 | | 0.92* | (0.84–1.01) | 0.92* | (0.83–1.01) |
| **Birth order** | | | | | | | | |
| | 1 child | 8,739 | 8.6 | **<0.001** | 1.0 | ref | 1.0 | ref |
| | 2–3 | 12,037 | 9.1 | | 1.05 | (0.93–1.19) | 0.94 | (0.83–1.07) |
| | 4–5 | 4,918 | 13.0 | | 1.58*** | (1.37–1.81) | 1.14 | (0.95–1.38) |
| | 6+ | 3,477 | 13.3 | | 1.62*** | (1.39–1.90) | 0.97 | (0.77–1.22) |
| **Birth weight (kg)** | | | | | | | | |
| | ≥ 2.5 Kg | 14,015 | 7.4 | **<0.001** | 1.0 | ref | 1.0 | ref |
| | < 2.5 Kg | 15,156 | 12.6 | | 1.82*** | (1.61–2.05) | 0.87* | (0.74–1.01) |
| **Place of delivered** | | | | | | | | |
| | Public | 10,072 | 6.7 | **<0.001** | 1.0 | ref | 1.0 | ref |
| | Private | 2,261 | 6.0 | | 0.90 | (0.69–1.18) | 1.05 | (0.79–1.39) |
| | At Home | 16,823 | 12.7 | | 2.04*** | (1.78–2.33) | 1.11 | (0.93–1.33) |
| **Breastfeeding** | | | | | | | | |
| | No | 13,704 | 9.3 | **0.003** | 1.0 | ref | 1.0 | ref |
| | Yes | 15,467 | 10.8 | | 1.17*** | (1.06–1.30) | **0.87**** | **(0.77–0.98)** |
| **BCG-vaccinated** | | | | | | | | |
| | None | 4,395 | 14.4 | **<0.001** | 1.0 | ref | 1.0 | ref |
| | Yes | 24,776 | 9.3 | | 0.61*** | (0.54–0.70) | 1.02 | (0.89–1.18) |

(*Continued*)

**Table 3.** (Continued)

| Variables | | 2000–2014 (N = 29,171) | | | 2000–2014 (N = 29,171) | | 2000–2014 (N = 29,115) | |
|---|---|---|---|---|---|---|---|---|
| | | Total | % ARI | P value | OR | 95% CI | AOR | 95% CI |
| **Intake vitamin A last 6 months** | | | | | | | | |
| | No | 15,289 | 11.3 | <0.001 | 1.0 | ref | 1.0 | ref |
| | Yes | 13,845 | 8.7 | | 0.75*** | (0.67–0.83) | 1.04 | (0.93–1.17) |
| **Place of residence** | | | | | | | | |
| | Urban | 4,176 | 8.1 | 0.002 | 1.0 | ref | 1.0 | ref |
| | Rural | 24,996 | 10.4 | | 1.33*** | (1.10–1.59) | 1.04 | (0.86–1.24) |
| **Wealth index** | | | | | | | | |
| | Poorest | 7,507 | 11.5 | <0.001 | 1.0 | ref | 1.0 | ref |
| | Poorer | 6,336 | 10.9 | | 0.94 | (0.82–1.07) | 0.93 | (0.82–1.07) |
| | Middle | 5,455 | 10.3 | | 0.88 | (0.76–1.03) | 0.89 | (0.76–1.04) |
| | Richer | 5,081 | 10.0 | | 0.86* | (0.72–1.01) | 0.90 | (0.75–1.07) |
| | Richest | 4,792 | 6.8 | | 0.57*** | (0.46–0.69) | **0.73**** | **(0.56–0.95)** |
| **Toilet facility** | | | | | | | | |
| | Improved | 8,521 | 6.0 | <0.001 | 1.0 | ref | 1.0 | ref |
| | Non-Improved | 20,649 | 11.8 | | 2.09*** | (1.80–2.42) | **1.20*** | **(0.99–1.46)** |
| **Drinking water** | | | | | | | | |
| | Improved | 16,005 | 10.4 | 0.282 | 1.0 | ref | - | - |
| | Unimproved | 13,160 | 9.8 | | 0.94 | (0.83–1.06) | - | - |
| **Survey years** | | | | | | | | |
| | 2000 | 7,284 | 19.9 | <0.001 | 1.0 | ref | 1.0 | ref |
| | 2005 | 7,201 | 8.6 | | 0.38*** | (0.32–0.44) | **0.36**** | **(0.31–0.42)** |
| | 2010 | 7,730 | 6.4 | | 0.28*** | (0.23–0.33) | **0.27**** | **(0.22–0.33)** |
| | 2014 | 6,956 | 5.5 | | 0.24*** | (0.20–0.28) | **0.24**** | **(0.19–0.30)** |

*** p<0.01

** p<0.05

* p<0.1

for a child's nutrition and the health of their immune response by breastfeeding have been suggested, among others transfer of anti-idiotypic antibodies and lymphocytes [36]. Children whose mothers had a secondary or higher education were found to be less likely than children whose mothers had no education to experience ARI symptoms. These results are in line with findings from Kenya, Ethiopia, and Rwanda studies [11, 37, 38]. This finding might be explained by the fact that highly educated mothers might have access to books and more education program that help to better safeguard their children [13]. Additionally, this research showed that children in higher wealth quintile households had a lower risk of developing ARI symptoms. Consistent with studied conducted in Bangladesh [39]. Higher family income has been linked to better living conditions, better nutritional status, and access to healthcare services, all of which have a positive impact on children's health outcomes. Financial stress on parents has a variety of effects on children's health and susceptibility to disease. For example, undernutrition, impaired cognitive development, and a weakened immune system in children are all strongly associated with financial stress, which increases the risk of infectious diseases. Compared to children from poor families, those from financially stable families are more likely to enjoy safe and secure housing with greater access to health-promoting conditions [40]. Children who were gathered between 2005 2010, and 2014 had a lower risk of developing ARI symptoms. Confirmed by studied conducted in East and South-East Asian nations [41]. This

decrease in the burden of ARI in these developing regions is the result of both a decline in incidence brought on by socioeconomic development and higher living standards as well as a rise in access and quality of care [22]. While proportion of children age 12–23 months who have been fully vaccinated against all basic antigens lower at 40% to 67% between 2000–2005 and peaked at 79% in 2010, has declined to 73% in 2014, and slightly increased to 76% in 2021–22 [42]. This study also demonstrated that, compared to 2000, there have been notable advancements in reducing U5 mortality due to infection diseases, which has been reduced, and achieved some remarkable child health outcomes reported to Millennium Development Goals (MDGs), such as a significant decline in child mortality rates in Cambodia by 2015 [43].

## Limitations and strengths

The study has the number of limitations. First, the temporal associations between independent variables and ARI symptoms were not able to assess at the same times given that CDHS was a cross-sectional study. Second, the use of secondary data from four different surveys in our analysis limits the inclusion of some potential factors, which were found to be associated with ARI disease in previous studies. These included children malnutritional, antibiotics prescription, type of roofing material, household's types of cooking, season effect, mode of delivered and number of antenatal care visits during pregnant. Third, the use of self-report from mothers to define the presence of ARI symptoms make our analysis prone to information bias. However, this recall bias appears to be non-differential. Despite the limitations, the study has several strengths. First study to assess the trends of ARIs over the 15 years. Second used of nationally representative data and a large sample size. Third the study was compared the associated factors with ARIs among under 5 years children from data survey to another and it is hoped that these findings will help the Cambodia Ministry of Health's Child Health Unit as well as health promotion and social determinants to plan interventions which will contribute to the reduction of ARI symptoms.

## Conclusions

In conclusion, the significant decrease in the prevalence of ARI among children aged 0–59 months from 19.5% in 2000 to 5.2% in 2014 reflects a solid achievement of the global and Cambodia efforts to reduce under 5 years morbidity and mortality. This paper revealed that the smoking mother, children's age, and households which non-improved toilet facilities were potential risk factors for ARI symptoms in children under 5 years. However, children living in richest wealth families, older children, children having highly educated mothers, breastfed children were less likely to develop of ARI disease symptoms. Our findings suggest policymakers and stakeholders in the healthcare sector should launch targeted initiatives to address the issues with inadequate child healthcare, unfavorable environmental conditions, and childcare facilities. Government and child family programs must promote maternal education, particularly infant breastfeeding. The government ought to support maternal education and infant breastfeeding in the interest of early childhood care. However, these data were gathered in 2000, 2005, 2010, and 2014 and might no longer reflect the current situation of children in Cambodia. Further evaluation of the prevalence of ARI symptoms and its determinants from CDHS 2020–21 could be done when the data becomes publicly available.

## Acknowledgments

The authors would like to thank DHS-ICF, who approved the data used for this paper.

## Author Contributions

**Conceptualization:** Samnang Um.

**Data curation:** Samnang Um.

**Formal analysis:** Samnang Um.

**Investigation:** Samnang Um.

**Methodology:** Samnang Um, Darapheak Chau.

**Project administration:** Samnang Um.

**Software:** Samnang Um.

**Writing – original draft:** Samnang Um, Daraden Vang, Punleak Pin, Darapheak Chau.

**Writing – review & editing:** Samnang Um, Darapheak Chau.

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
