## [Decision Letter · Decision Letter 0]

14 Feb 2023

PGPH-D-22-01879

Trends and determinants of Acute Respiratory Infection symptoms among Under-five children in Cambodia: Analysis of 2000 to 2014 Cambodia Demographic and Health Surveys

Dear Dr. Samnang Um,

Thank you for submitting your manuscript to PLOS Global Public Health. After careful consideration, we feel that it has merit but does not fully meet PLOS Global Public Health’s publication criteria as it currently stands. Therefore, we invite you to submit a revised version of the manuscript that addresses the points raised during the review process.

We look forward to receiving your revised manuscript.

Kind regards,

Atul Vashist, PhD

Academic Editor

Journal Requirements:

1. PLOS Global Public Health does not copy edit accepted manuscripts (https://journals.plos.org/globalpublichealth/s/criteria-for-publication#loc-5). To that effect, please ensure that your submission is free of typos and grammatical errors.

2. We have noticed that you have uploaded Supporting Information files, but you have not included a list of legends. Please add a full list of legends for your Supporting Information files after the references list. 

3. In the online submission form, you indicated that your data will be submitted to a repository upon acceptance.  We strongly recommend all authors deposit their data before acceptance, as the process can be lengthy and hold up publication timelines. Please note that, though access restrictions are acceptable now, your entire data will need to be made freely accessible if your manuscript is accepted for publication. This policy applies to all data except where public deposition would breach compliance with the protocol approved by your research ethics board. If you are unable to adhere to our open data policy, please kindly revise your statement to explain your reasoning and we will seek the editor's input on an exemption. Please be assured that, once you have provided your new statement, the assessment of your exemption will not hold up the peer review process.

Additional Editor Comments (if provided):

Reviewers' comments:

Reviewer's Responses to Questions

**Comments to the Author**

1. Does this manuscript meet PLOS Global Public Health’s publication criteria? Is the manuscript technically sound, and do the data support the conclusions? The manuscript must describe methodologically and ethically rigorous research with conclusions that are appropriately drawn based on the data presented.

Reviewer #1: Partly

Reviewer #2: Yes

2. Has the statistical analysis been performed appropriately and rigorously?

Reviewer #1: Yes

Reviewer #2: No

3. Have the authors made all data underlying the findings in their manuscript fully available (please refer to the Data Availability Statement at the start of the manuscript PDF file)?

Reviewer #1: Yes

Reviewer #2: Yes

4. Is the manuscript presented in an intelligible fashion and written in standard English?

Reviewer #1: Yes

Reviewer #2: Yes

5. Review Comments to the Author

Reviewer #1: Manuscript is well written and respects scientific norms. It is also of public health health importance but as mentioned by the authors under limitations, there are worries about the relevance of how the conclusions of the research could be exploited because probably they might not reflect what current epidemiological data would reveal.

Furthermore, authors need to check the interpretation of AOR"s and the confidence intervals so that the comments tally with the variables analyzed.

The grammar also needs to checked. Please, see the attached tracked copy for comments and edits.

Reviewer #2: In this manuscript titled “Trends and determinants of Acute Respiratory Infection symptoms among Under-five children in Cambodia: Analysis of 2000 to 2014 Cambodia Demographic and Health Surveys” the authors have comprehensively discussed about the scenario and prevalence of acute respiratory infection symptoms among the children of age below 5 years. Though the authors have designed the study in an organised way and the sample number of the study is also good, but I have some major concerns listed below.

1. The data presented in the study are very old from 2000-2014. There is no much information about the recent data of the acute respiratory infections in Cambodia. So, I suggest to include the updated information.

2. Also, the authors should mention the previous studies which linked the prevalence of acute respiratory infections with unhygienic toilet in the discussion section

3. It would also be of importance if the authors can include the data of intake of any antibiotics in the last 6 months, which will improve the significance of the study.

4. Kindly recheck the reference part of the manuscript, it is not in the systematic order.

5. Also, the reduction in the children death may be due to the robust vaccination program, which could be highlighted in the manuscript.

6. PLOS authors have the option to publish the peer review history of their article (what does this mean?). If published, this will include your full peer review and any attached files.

**Do you want your identity to be public for this peer review?** For information about this choice, including consent withdrawal, please see our Privacy Policy.

Reviewer #1: **Yes: **Professor Gregory Edie Halle-Ekane (MD, FWACS)

Reviewer #2: **Yes: **Dr. Deepjyoti Paul

---

## [Decision Letter · Decision Letter 1]

3 Apr 2023

Trends and determinants of Acute Respiratory Infection symptoms among Under-five children in Cambodia: Analysis of 2000 to 2014 Cambodia Demographic and Health Surveys

PGPH-D-22-01879R1

Dear Dr. Samnang Um,

We are pleased to inform you that your manuscript 'Trends and determinants of Acute Respiratory Infection symptoms among Under-five children in Cambodia: Analysis of 2000 to 2014 Cambodia Demographic and Health Surveys' has been provisionally accepted for publication in PLOS Global Public Health.

Best regards,

Atul Vashist, PhD

Academic Editor

Reviewer Comments (if any, and for reference):

Reviewer's Responses to Questions

**Comments to the Author**

1. If the authors have adequately addressed your comments raised in a previous round of review and you feel that this manuscript is now acceptable for publication, you may indicate that here to bypass the “Comments to the Author” section, enter your conflict of interest statement in the “Confidential to Editor” section, and submit your "Accept" recommendation.

Reviewer #1: All comments have been addressed

Reviewer #2: All comments have been addressed

2. Does this manuscript meet PLOS Global Public Health’s publication criteria? Is the manuscript technically sound, and do the data support the conclusions? The manuscript must describe methodologically and ethically rigorous research with conclusions that are appropriately drawn based on the data presented.

Reviewer #1: Yes

Reviewer #2: Yes

3. Has the statistical analysis been performed appropriately and rigorously?

Reviewer #1: Yes

Reviewer #2: Yes

4. Have the authors made all data underlying the findings in their manuscript fully available (please refer to the Data Availability Statement at the start of the manuscript PDF file)?

Reviewer #1: Yes

Reviewer #2: Yes

5. Is the manuscript presented in an intelligible fashion and written in standard English?

Reviewer #1: Yes

Reviewer #2: Yes

6. Review Comments to the Author

Reviewer #1: Good.Comments made have been adequately addressed especially the statistical analysis, grammar and discussion.

The justification given as regards the validity of data compared to the present epidemiological variables seems acceptable.

Reviewer #2: (No Response)

7. PLOS authors have the option to publish the peer review history of their article (what does this mean?). If published, this will include your full peer review and any attached files.

**Do you want your identity to be public for this peer review?** For information about this choice, including consent withdrawal, please see our Privacy Policy.

Reviewer #1: **Yes: **Professor Halle-ekane Gregory MBBS; MD; FWACS

Reviewer #2: **Yes: **Dr. Deepjyoti Paul
